# Clinical efficacy and effectiveness of 3D printing: a systematic review

Laura E Diment, Mark S Thompson, Jeroen H M Bergmann

Department of Engineering Science, University of Oxford, Oxford, UK

**Correspondence to**
Dr Jeroen H M Bergmann;
jeroen.bergmann@eng.ox.ac.uk

## ABSTRACT

**Objective** To evaluate the clinical efficacy and effectiveness of using 3D printing to develop medical devices across all medical fields.

**Design** Systematic review compliant with Preferred Reporting Items for Systematic Reviews and Meta-Analyses.

**Data sources** PubMed, Web of Science, OVID, IEEE Xplore and Google Scholar.

**Methods** A double-blinded review method was used to select all abstracts up to January 2017 that reported on clinical trials of a three-dimensional (3D)-printed medical device. The studies were ranked according to their level of evidence, divided into medical fields based on the International Classification of Diseases chapter divisions and categorised into whether they were used for preoperative planning, aiding surgery or therapy. The Downs and Black Quality Index critical appraisal tool was used to assess the quality of reporting, external validity, risk of bias, risk of confounding and power of each study.

**Results** Of the 3084 abstracts screened, 350 studies met the inclusion criteria. Oral and maxillofacial surgery contained 58.3% of studies, and 23.7% covered the musculoskeletal system. Only 21 studies were randomised controlled trials (RCTs), and all fitted within these two fields. The majority of RCTs were 3D-printed anatomical models for preoperative planning and guides for aiding surgery. The main benefits of these devices were decreased surgical operation times and increased surgical accuracy.

**Conclusions** All medical fields that assessed 3D-printed devices concluded that they were clinically effective. The fields that most rigorously assessed 3D-printed devices were oral and maxillofacial surgery and the musculoskeletal system, both of which concluded that the 3D-printed devices outperformed their conventional comparators. However, the efficacy and effectiveness of 3D-printed devices remain undetermined for the majority of medical fields. 3D-printed devices can play an important role in healthcare, but more rigorous and long-term assessments are needed to determine if 3D-printed devices are clinically relevant before they become part of standard clinical practice.

### Strengths and limitations of this study

► This is the first rigorous systematic literature review of three-dimensional printing for clinical uses.
► Validated quality assessment and clinical level of evidence tools are used to assess the progress made in different medical fields.
► The study is limited to a critical appraisal of individual studies, rather than a meta-analysis, because of the breadth of uses (from anatomical models and surgical guides to therapeutic devices) and the lack of comparable hypotheses.
► Due to the speed of innovation in the field, the review will need to be updated frequently.

## INTRODUCTION

Three-dimensional (3D) printing is likely to play a pivotal role in transforming healthcare and clinical practice because it provides the opportunity to create customised devices designed for the complexity and individual variances of the patient populations.[1]

Additive manufacturing, more commonly referred to as 3D printing, is an industrial production technique that enables a 3D digital model to be converted into a physical model by printing it layer by layer. For decades, 3D printing has been used for rapid prototyping, but the recent advances in the available materials, speed, resolution, accuracy, reliability, cost and repeatability of 3D printing technologies have broadened the possibilities for clinical uses.[2 3] Many medical fields are already using 3D printing to manufacture custom surgical tools, guides, dose delivery devices, implants, external prosthetics or orthotics and devices for preoperative planning or education.[2 4 5] Tailoring devices and procedures to the patient are expected to reduce the time required for surgery, treatment or recovery, while increasing the accuracy and success of the outcome.[1] The worldwide 3D printing industry's revenue from products and services is over US$4 billion and fast growing, with 13.1% of the industry attributed to the medical sector.[6]

In most medical fields, 3D printing applications are still in the research and development stage or have only just entered clinical practice within the last decade, and hence, there has been a lack of research into the clinical efficacy, effectiveness and long-term follow-up in comparison to traditional technologies.[7] *Efficacy* refers to the performance of the device under ideal and controlled conditions, and

*effectiveness* refers to its performance under typical clinical conditions.[8] The Food and Drug Administration states that 3D-printed medical devices are required to meet the same regulations as their non–3D-printed counterparts. Because 3D-printed devices can have different safety and efficacy issues than the equivalent devices, additional testing may be required to demonstrate safety, efficacy and effectiveness.[9] Hospitals and clinics will need to adopt new medical product procedures when they want to introduce 3D printing for healthcare, and more evidence of device efficacy and effectiveness will help make an informed discussion before prescribing 3D-printed devices for patients. Providing a critical appraisal of the efficacy and effectiveness of 3D-printed medical devices gives healthcare professionals a resource to assess the validity of the devices and provides researchers with an overview of areas that require further research and validation.

Previous systematic reviews summarise 3D-printed medical devices being used in specific medical fields, such as plastic and reconstructive surgery[10] and preoperative planning for liver resections,[11] and increasingly available 3D printing processes used in dentistry and mandibular reconstructions.[12–14] Systematic reviews on the advantages of 3D-printed devices over conventional methods in surgery have found improved clinical outcomes and reductions in operating times and manufacturing costs.[5 15] However, the reviews do not critically appraise the studies and are therefore subject to bias. The only systematic review that did assess bias for a very specific field found no difference in patient outcomes when 3D-printed instrumentation for total knee replacements was compared with conventional instrumentation.[16] This highlights the need for a systematic review that incorporates a critical appraisal of the studies that are included.

This review aims to assess the clinical efficacy and effectiveness of 3D-printed devices through performing a systematic literature search, categorising reports of 3D-printed device usage by medical field and purpose and assessing the reports' scientific quality. The key findings of studies that compare 3D-printed medical devices with their non–3D-printed counterparts are presented, and the gaps in research were highlighted.

## METHODS
### Search strategy
The applied protocol (not previously registered) used systematic methods to search for relevant studies, screen them for eligibility and assess their quality. The review follows the Preferred Reporting Items for Systematic Reviews and Meta-Analyses guidelines.[17] A literature search was performed using PubMed, Web of Science, OVID, IEEE Xplore and Google Scholar. A combination of 45 relevant keywords were used to collect all studies that included a 3D-printed device and a clinical trial. Due to the word and character limit search restrictions in IEEE Xplore and Google Scholar, a narrower search of 15 terms was performed for these databases. The search includes all publications up to January 2017. The complete search strategy is provided in the online supplementary material. Mendeley was used as a reference manager.

### Study selection
After removing all duplicates from the databases, the title and abstract of each publication was reviewed using a double-blinded method (undertaken by LED and JHMB) to determine eligibility for inclusion.

The inclusion criteria were:

i. Relevance: papers were required to report first-hand on the results of a clinical study that assessed the efficacy and effectiveness of a 3D-printed device.
ii. Language: only papers written or translated into English were included in the review.
iii. Peer review: records that had no peer review or where the level of peer review could not be traced were excluded from the review.

Where the results of the reviewers (LED and JHMB) conflicted, the reviewers discussed their reasoning for inclusion/exclusion, and where required for clarification of the paper's relevance, the paper's introduction and methods were assessed. Consensus on eligibility was achieved for all papers. Included articles underwent a full-text review, and references cited in these studies were also examined for relevance using the same inclusion criteria.

### Quality assessment
The studies were then rated according to their level of evidence, based on the Oxford Centre for Evidence-based Medicine Levels of Evidence,[18] as summarised in table 1. Systematic reviews and opinion papers were excluded because they cannot provide the original data in a suitable format to be assessed using the quality assessment method used in this review. Where studies used 3D-printed devices but the hypothesis, aim or objective was focused on an outcome other than the effect of the 3D-printed device, the paper has been graded as the level of evidence shown for the device. All levels of evidence were included in the review, but only those that used a control group (levels 1–3) were critically analysed.

**Table 1** Centre for Evidence-based Medicine Levels of Evidence

| Level | Types of study |
|---|---|
| 1 | Randomised controlled trials |
| 2 | Cohort studies |
| 3 | Case–control studies |
| | Poor quality estimates of data that include sensitivity analyses incorporating clinically sensible variations |
| 4 | Case-series studies |
| 5 | Case reports |
| | Studies that do not provide quantitative evidence to support a hypothesis |

Studies were then placed in medical fields based on the International Classification of Diseases chapter divisions[19] and categorised by the purpose of the device. The three categories were devices used for preoperative planning, devices to aid surgery and therapeutic devices.

To analyse the quality of the studies under review, critical appraisal tools were assessed to find one that met the following criteria:

Suitable for assessing the quality of both randomised and non-randomised studies,

Well regarded and commonly used for quality assessments in systematic reviews.

Demonstrates internal consistency, test–retest reliability and criterion validity.

Assesses the risk of bias, the participant selection methods, the study protocol and the validity, reliability and responsiveness of the study's results.

Simple and intuitive to interpret.

Does not assign weights to the items in the scale where there is a lack of empirical evidence to support the assignment.[20]

The Downs and Black Quality Index[21] meets all but the final criterion on this list. To overcome this limitation, responses have been left in their raw form (yes/no/unable to determine/not applicable, in accordance with the definitions provided by Downs and Black[21]) and presented in a table to enable the reader to visualise the category trends, as recommended by the Cochrane Handbook for Systematic Reviews of Interventions.[20]

The study details and data relating to the quality of study design and reporting were manually extracted from each paper by one reviewer (LED). The study details included medical field, study design, device category, device purpose, number of participants, age and gender of participants and aims and outcomes of the study. The data relating to study design and reporting were extracted according to the criteria used by the Downs and Black Quality Index, divided into reporting, external validity, bias, confounding and power. Where there were uncertainties as to how the study fitted with the Downs and Black Quality Index criteria, the other two authors were consulted, and in all cases, an agreement was reached.

## RESULTS

The search yielded 4505 records, and after removing duplicates, 3084 abstracts were screened. A total of 350 studies met the inclusion criteria and were included in the review (figure 1).

Due to the large number of included studies, this paper gives an overview of the level of evidence found for 3D-printed medical devices in each medical field and then focuses on the outcomes of the level 1 studies. Information for all studies included in the review and a critical appraisal of all comparative studies is provided in online supplementary material.

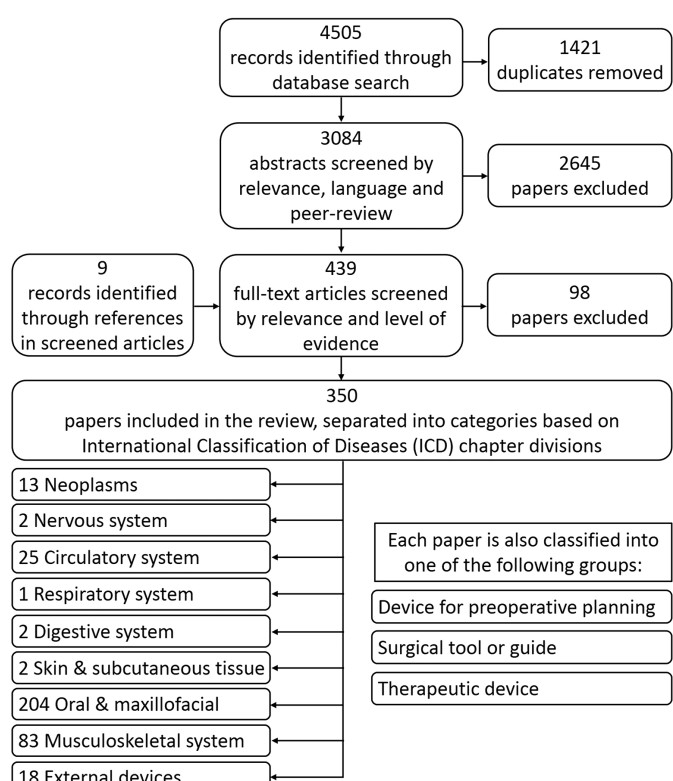

**Figure 1** Flow chart of the selection and sorting method.

A wide range of 3D-printed devices have been developed and clinically trialled. The most common device types were anatomical models for preplanning surgeries (40.9%), followed by guides to aid in surgery (37.1%). The studies covered nine medical fields, with the majority (58.3%) of studies falling into the oral and maxillofacial surgery field, which included dentistry and orthopaedic surgery of the jaw, face and skull, and those covering the musculoskeletal system (23.7%) making up the second group. Some studies spanned multiple medical fields, such as neoplasms that occur in the circulatory system. In these cases, the authors selected the most prominent field of the paper. Only 14.0% of included studies used a control group (levels 1–3 in figure 2), whereas 41.4% were level 5 studies. The average number of participants per controlled trial was 41 (36 for randomised controlled trials (RCTs)), compared with 17 for all studies.

All 21 level 1 studies fell within the oral and maxillofacial surgery or musculoskeletal system categories, with one cross-field study fitting in both oral and maxillofacial surgery and neoplasms (table 2 summarises all RCTs). All level 1 studies, with the exception of Stephens et al,[22] used objective measures as their indicators of device efficacy and effectiveness. Of these, the seven models used for preoperative planning were all modelling bone or fracture fixation plates, though the locations spanned skull, face, mandible, arm, spine, hip and ankle. All aimed to assess the effectiveness of using the models for surgical planning in treating specific conditions. Six out of seven studies used operating time, four used accuracy of the surgery and three used blood loss as key indicators of

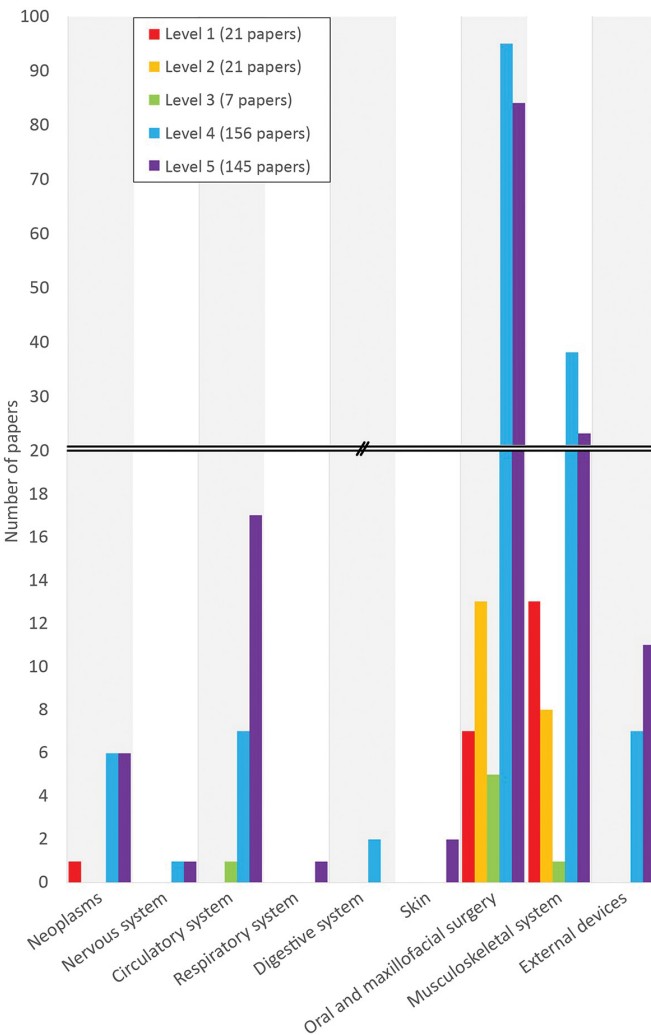

**Figure 2** Number of studies per level of evidence given by medical field. Levels of evidence are colour coded and fields are visually separated by alternating between white and grey columns.

device effectiveness. Aesthetics, recovery time and post-surgery function were also used as indicators of device effectiveness. All level 1 studies of 3D-printed models for preoperative planning that used time, accuracy or blood loss as indicators of device effectiveness reported that the test group had better outcomes than the control group, with statistical significance at P<0.05 found for four studies that assessed operating time, three studies that assessed accuracy and two studies that assessed blood loss. The two studies that assessed recovery time found no difference between groups, one of the two studies that assessed user function found statistically significant differences in the test group and the other found no difference between groups.

Of the 11 RCTs that assessed 3D-printed guides or templates that supported navigation and surgery (of the mandible, teeth, shoulder, spine, hip or knee), 9 studies used accuracy of positioning as an indicator of device effectiveness, all of which found an improvement when using the guide or template and 8 of which showed

statistically significant results at P<0.05. All four studies that used time as an indicator of device effectiveness showed decreased operating times when using the guide or template (all with P<0.05), and five of the six studies that used changes at the operation site (blood loss, inflammatory response, tactile sensation or aesthetics) found statistically significant improvements at P<0.05.

Only three therapeutic devices were tested using RCTs. These were zirconia crowns, which were found to have a better marginal gap and internal fit than the conventional crowns tested (P<0.05); cranial implants, which showed less vertical ridge resorption over time (P<0.05); and osteosynthesis plates for intercondylar humeral fractures, which showed a reduction in operating time (P<0.05) but no difference in elbow function during recovery.

While most level 1 studies found statistically significant improvements at P<0.05 when using the 3D-printed device, none performed an analysis on the clinical relevance of these improvements.

Critical appraisal of the RCTs from each medical field is shown in table 3. The table includes the abridged quality index questions. Studies with higher levels of evidence typically rated better across all areas assessed using the quality index. However, many of the RCTs did not demonstrate external validity and did not blind participants or those measuring the outcomes or assess confounding factors between groups. Only 57% of the RCTs included follow-up.

## DISCUSSION

The results show that a wide range of 3D-printed devices have been clinically trialled, but few papers have rigorously assessed the efficacy or effectiveness of clinical 3D-printed devices. The 3D-printed devices tested were mainly designed for planning surgeries, particularly using anatomical models or surgical guides to aid cutting or navigating, with little attention dedicated to 3D-printed interventions. This is likely due to the lower complexity of the anatomical models and surgical guides and the smaller risk to the patient than therapeutic devices, resulting in less rigorous regulation and safety testing requirements. Approximately twice as many studies were in oral and maxillofacial surgery than in the musculoskeletal system, the two largest fields, but the musculoskeletal system, despite its more recent uptake of 3D printing, had almost double the number of level 1 studies. This is perhaps because clinical trials in musculoskeletal surgery typically aim to demonstrate quantitative clinical outcomes, driving more rapid growth in the level of evidence of studies than oral and maxillofacial surgery where aesthetics is a priority and individual outcomes are harder to quantify.

The RCT listed under neoplasms was an anatomical model used for planning mandibular surgery, but with a focus on tumour removal, fitting into both categories. Therefore, no RCTs were found outside the oral and maxillofacial surgery or musculoskeletal system fields. The primary indicators of

**Table 2** Summary of randomised controlled trials (type: p=preoperative planning, S=surgical tool or guide, T=therapeutic device)

| Sector | First author, date | Device | Type | Size (n) | Participant details | Aim | Outcomes |
|---|---|---|---|---|---|---|---|
| Neoplasms | de Farias, 2014[27] | Mandible model | P | 37 (test=17, control=20) | Age: 9–74 years (test mean=39.57 years, control mean=53.11 years). Test=14M/3F, control=15M/5F | Aim: to evaluate the efficacy of using 3D-printed models to plan head and neck surgeries. | Reconstruction time decreased (test=43.7 min, control=127.7 min, P=0.001). Though statistical significance was not demonstrated, the size of the bone flap taken for reconstruction was typically smaller, and the aesthetic results better in the test group. |
| Oral and maxillofacial | Ahrberg, 2015[28] | All-ceramic zirconia crowns | T | 25 (same participants used for test and control) | Age: >18 years 10M/15F | Aim: to evaluate the marginal and internal fit of 3D-printed zirconia crowns and three-unit fixed dental prostheses resulting from direct versus indirect digitalisation. | The marginal gap was smaller (test=61.08±24.77 µm, control=70.40±28.87 µm, P<0.05), and the fit was better at the centro-occlusal site (test=155.57±49.85, control=171.51±60.98, P<0.05). The fit improved at the mid-axial wall and axio-occlusal transition, but the results lacked statistical significance. |
| Oral and maxillofacial | D'Urso, 1999[29] | Craniomaxillofacial model | P | 30 (test=15, control=15) | No information provided | Aim: to assess whether biomodels in addition to standard imaging have greater utility in patient education, diagnosis and operative planning than the standard imaging alone. | 3D-printed models improved operative planning accuracy (test=82.21%, control=44.09%, P<0.01) and diagnosis accuracy (test=95.23%, control=65.63%, P<0.01). Measurement accuracy improved (test error=7.91%, control error=44.14%, P<0.05). Surgeons estimated that operating time over 45 operations was also reduced by 17.63%. |
| Oral and maxillofacial | Al-Ahmad, 2013[30] | Surgical guide for sagittal splitting ramus osteotomy | S | 8 (one side used as test, the other as control) | Age: 18–30 years (mean=23 years) | Aim: to evaluate the effectiveness of sagittal splitting ramus osteotomy in reducing the incidence and severity of neurosensory alterations, using a 3D-printed surgical guide. | The Semmes-Weinstein monofilament tactile threshold was considered abnormal at >2.83 on the $\chi^2$ test. At 1 week after surgery: test=67% abnormal, control=83% abnormal. This difference showed statistical significance (P<0.05) at 3 months postsurgery for chin and 6 months postsurgery for lower lip. For the two-point discrimination, the lower lip showed statistical significance at 1 week (test=9 mm, control=22 mm) and the chin at 6 months (test=11 mm, control=30 mm). |
| Oral and maxillofacial | Ayoub, 2014[31] | Guide for mandibular reconstruction | S | 20 (test=10, control=10) | Age: test=52.3±19.2 years, control=54.7±14.2 years. Test=4M/6F, control=6M/4F | Aim: to evaluate the benefits of computer-assisted mandibular reconstruction with iliac crest bone grafts regarding the intraoperative time for transplant shaping, ischaemia, duration of surgery, amount of bone removed and the change in postoperative condyle position compared with conventional surgery. | Computer-assisted surgery shortened the time of transplant ischaemia (test=96.1±15.8 min, control=122.9±20.4 min, P<0.005) and defect reconstruction (test=6.2±4.9 min, control=20.3±7.4 min, P<0.001). There was less bone harvested: test=no difference regarding defect size, control=16.8±5.6 mm larger transplant size than defect size, P<0.001. The intercondylar distance before compared with after surgery was less affected: test=1.3±0.2 mm, control=5.5±2.5 mm, P<0.001. |
| Oral and maxillofacial | Goh, 2015[32] | Polycaprolactone scaffold for ridge preservation | T | 13 (test=6, control=7) | Age: 29–60 years (mean=46.8 years). 7M/7F (1 lost to follow-up, so excluded from results) | Aim: to evaluate the feasibility and effectiveness of using a polycaprolactone scaffold in fresh extraction sockets for ridge preservation. | The test group showed less vertical ridge resorption in all aspects (mesio-buccal, mid-buccal and disto-buccal) with statistically significant results in the mesio-buccal aspect (test=0.13±0.96 mm, control=−2.18±1.02 mm, P=0.008), with better maintenance of ridge height after 6 months. |
| Oral and maxillofacial | Van de Velde, 2010[33] | Maxilla surgical guide | S | 13 (test=36 tooth implants, control=34 tooth implants) | Age: 39–75 years (mean=55.7 years), 4M/9F | Aim: to compare the outcome of dental implants placed using a flapless protocol with a 3D-printed guide and immediate loading with a conventional protocol and loading. | The height of the attached mucosa at 1 week postsurgery was less (test=3.26±1.57 mm, control=6.01±1.10 mm, P<0.05). Marginal bone levels were not statistically significantly different between test and control implants (test=1.95 mm±0.70 and 1.93 mm±0.42 after 18 months). Opinion about speech, function, aesthetics and self-confidence was statistically significantly better for the test side after 1 week when using a VAS rating scale, P<0.05. |
| Oral and maxillofacial | Vercruyssen, 2014[34] | Surgical guides for edentulous jaws | S | 59 (72 jaws, 12 in each group) | Age: 31–78 years 31 male jaws, 41 female jaws | Aim: to assess the accuracy of guided surgery compared with mental navigation or the use of a surgical template, in fully edentulous jaws. | There was lower deviation at the entry point (1.4 mm), at the apex (1.6 mm) and angular deviation (3.0°) when using the 3D-printed guiding systems (all P<0.05). There were negligible differences between bone and mucosa support or type of guidance. |
| Musculoskeletal | Chareancholvanich, 2013[23] | Cutting guides for total knee replacement | S | 80 (test=40, control=40) | Age: 55–84 years (mean=69.5 years) 6M/34F | Aim: to compare patient-specific cutting guides with conventional instrumentation in total knee replacement. | The tibial component in the test group was marginally closer to neutral alignment (test=89.8±1.2°, control=90.5±1.6°, P=0.03). The 3D-printed guides also shortened the bone-cutting time by 3.6 min (P<0.001) and the operating time by 5.1 min (P=0.019), without differences in postoperative blood loss (P=0.528) or need for blood transfusion (P=0.789). These minimal advantages of the 3D-printed guides are unlikely to be clinically relevant. |

Continued

**Table 2** Continued

| Sector | First author, date | Device | Type | Size (n) | Participant details | Aim | Outcomes |
|---|---|---|---|---|---|---|---|
| Musculoskeletal | Chen, 2015[35] | Guide plate for pedicle screw fixation | S | 43 (test=20, control=23) | Age: test=35–70 years (mean=52.3 years), control=37–72 years (mean=55.4 years). Test=9M/11F, control=12M/11F | Aim: to evaluate the clinical efficacy of use of a 3D printing guide plate in posterior lumbar pedicle screw fixation. | Placement time for each screw was shorter (test=4.9±2.1 min, control=6.5±2.2 min), the amount of haemorrhage was less (test=8.0±11.1 mL, control=59.9±13.0 mL) and the fluoroscopy times of each screw placement was lower (test=0.5±0.4 min, control=1.2±0.7 min), $P<0.05$ for all tests. The excellent and good screw placement rate was 100% in the test group and 98.4% in the control group, with no statistical difference. No complications were reported. |
| Musculoskeletal | Du, 2013[36] | Templates to aid in pin placement in hip resurfacing arthroplasty | S | 34 (test=16, control=18) | Ages: 37–55 years | Aim: to assess whether 3D-printed patient-specific templates aid in accurate intraoperative pin placement. | The prosthesis stem shaft angle was greater (test=138.68±8.85°, control=118.9±12.8, $P=0.001$). The locating template can provide precise and dependable location for hip resurfacing femoral components during arthroplasty and ensure the valgus stem placement necessary for optimal outcomes. |
| Musculoskeletal | Gan, 2015[37] | Navigational template for total knee arthroplasty | S | 70 (test=35, control=35) | Age: Test=63–75 years (mean=68 years), control=64–72 years (mean=67.8 years). Test=10M/25F, control=9M/26F | Aim: to validates a novel patient-specific navigational template for total knee arthroplasty. | The 3D-printed navigational template reduced operation time (test=45±8 min, control=60±10 min, $P<0.001$) and blood loss (test=200±45 mL, control=290±60 mL, $P<0.001$). The mean deviation from the neutral axis also decreased for the frontal femoral component (test=1.0±0.8°, control=2.6±1.8°, $P<0.001$) and the frontal tibial component (test=1.2±1.0°, control=2.8±1.5°, $P<0.001$). |
| Musculoskeletal | Hendel, 2012[38] | Guide for determining glenoid component position | S | 31 (test=15, control=16) | No information provided | Aim: to compare patient-specific instruments with standard surgical instruments in determining glenoid component position. | The 3D printing technology decreased the average deviation of implant position for inclination (test=2.9±3.4°, control=11.6±7.0°, $P<0.0001$) and medial-lateral offset (test=1.0±0.9 mm, control=1.9±1.0 mm, $P<0.05$). The greatest benefit was observed in patients with retroversion >16° (test=1.2±2.0°, control=10±4.4°, $P<0.001$). |
| Musculoskeletal | Maini, 2016[39] | Plate for acetabulum fracture fixation | P | 21 (test=10, control=11) | Age: 18–60 years (mean=38.7 years) Test=9M/1F, control=9M/2F | Aim: to evaluate the accuracy of patient-specific precontoured plates. | Greater reduction was achieved, as evaluated by CT scan (test=8.36±5.82 mm, control=2.99±1.34 mm, $P<0.05$). Reduced blood loss (test=620±247 mL, control=720±286 mL) and surgical time (test=120±38 min, control=132±41 min) were observed, but the results were not statistically significant ($P>0.05$). All plates fitted well to the pelvis intraoperatively. |
| Musculoskeletal | Merc, 2013[40] | Drill guide for screw placement | S | 19 (test=9 (54 screws), control=10 (54 screws)) | Age: test=59±5 years, control=62±12 years. Test=4M/5F, control=5M/5F | Aim: to evaluate the accuracy of a multi-level drill guide template for lumbar and first sacral pedicle screw placement and to compare it with the freehand technique under fluoroscopy supervision. | The incidence of cortex perforation was reduced (test=0, control=8, $P<0.05$). So was the deviation of pedicle screw position in the sagittal plane (test=2±10°, control=−12±8°, $P<0.05$), but there was no discernable difference in the transverse plane deviation or displacement in sagittal or transverse planes. |
| Musculoskeletal | Shuang, 2016[41] | Osteosynthesis plates | T | 13 (test=6, control=7) | Age: test=31–62 years (mean=46.2 years), control=27–59 years (mean=40.3 years). Test=4M/2F, control=6M/1F | Aim: to evaluate the efficacy custom 3D-printed osteosynthesis plates in the treatment of intercondylar humeral fractures. | Operative time was shorter (test=70.6±12.1 min, control=92.3±17.4 min, $P<0.05$). The mean time to bone union was 3.4 months. No difference was found in the rate of patients with good elbow function or in the ranges of elbow flexion/extension and pronation/supination (all $P>0.05$). |
| Musculoskeletal | Stephens, 2016[22] | Bone models | P | 58 (test=29, control=29) | Age: >60 years (mean=73 years) 5M/53F | Aim: to investigate the efficacy of 3D-printed bone models as a tool to facilitate initiation of bisphosphonate treatment among individuals who are newly diagnosed with osteoporosis. | Using the nine-item Brief Illness Perception Questionnaire with a 0–10 rating scale, the test group was more emotionally affected by osteoporosis immediately after the interview (test=4.08±0.41, control=2.89±0.40, $P>0.05$) and reported a greater understanding of osteoporosis at the 2-month follow-up (test=7.19±0.51, control=5.72±0.49, $P<0.04$). |
| Musculoskeletal | Wu, 2011[42] | Spine model | P | 62 (test=34, control=28) 677 screws inserted | Age: 4–22 years (mean=11 years) | Aim: to compare the accuracy and safety of pedicle screw placement in congenital scoliosis using the 3D printing technique versus conventional fluoroscopy. | Results for thoracolumbar and lumbar, respectively: Shorter operation time (test=6.5 and 5.1 hours, control=8.5 and 6.3 hours, $P<0.05$). Higher scoliosis correction ratio (test=83.8 and 81.8, control=76.6 and 64.7, $P<0.05$). Accuracy of screw placement (test=94.4% and 91.6%, control=86.1% and 82.0%, $P<0.05$). |

Continued

**Table 2** Continued

| Sector | First author, date | Device | Type | Size (n) | Participant details | Aim | Outcomes |
|---|---|---|---|---|---|---|---|
| Musculoskeletal | Yang, 2016[43] | Model of trimalleolar fracture | P | 30 (test=15, control=15) | Age: 31–42 years (mean=36.5 years) 16M/14F | Aim: to evaluate the effectiveness of using 3D-printed models for surgical planning in treating trimalleolar fractures and in physician–patient communication. | Shorter operation tim; (test=184.32±4.65 min, control=212.32±8.17 min, P<0.001). Less blood loss (test=846.68±26.11 mL, control=1029.65±72.18 mL, P<0.001). No statistically significant differences were observed in complication rate, length of hospital stay and postoperative radiological outcomes (all p>0.05). |
| Musculoskeletal | You, 2016[44] | Proximal humeral fracture model | P | 66 (test=34, control=32) | Age: 61–76 years (test=66.09±4.09 years, control=66.28± 4.10 years) 27M/39F | Aim: to investigate the feasibility and clinical potential of using 3D-printed models versus the conventional thin-layer CT scan for the treatment of proximal humeral fractures in old people. | Surgery time decreased (rest=77.65±8.09 min, control=92.03±10.31 min, P<0.05), and there was less blood loss (test=235.29±63.40 mL, control=281.25±57.85 mL, P<0.05) and a lower number of fluoroscopy (test=7.12±1.57, control=10.59±1.36, P<0.05). The results showed no statistically significant difference in time to union (P>0.05). |
| Musculoskeletal | Zhang, 2011[45] | Acetabular navigational template | S | 22 (test=11, control=11) | Age: test=48.6±6.8 years, control=49.3±4.9 years. Test=7M/4F, control=5M/6F | Aim: to compare a customised 3D-printed acetabular navigational template for total hip arthroplasty to conventional THA in adult single development dysplasia of the hip. | The templates facilitated accurate placement of acetabular components in dysplasia of acetabulum. At 1-year follow-up, there were smaller differences from the predetermined angles of 45° abduction and 18° anteversion (test=1.6±0.4°, 1.9±1.1°, control=5.8±2.9°, 3.9±2.5°, P<0.05). |

THA, total hip arthroplasty; VAS, Visual Analogue Scale.

success used to determine the efficacy and effectiveness of 3D-printed devices across all medical fields were operating time, accuracy of surgery or positioning, fit, changes at the operation site (particularly blood loss, inflammation and aesthetics), recovery time and functional outcomes. From the studies that compared against a control, the 3D-printed devices used for preoperative planning and aiding surgery consistently found decreases in operating time and increases in surgical accuracy, the two most commonly reported indicators of effectiveness. Operating time is an obvious choice as an indicator for device success because it is easy to measure and quantify and corresponds to decreased blood loss and faster recovery. The main aims of anatomical models and surgical guides are to provide better information to the surgeon on the surgical site and to guide the surgeon's hand. Therefore, measuring the accuracy of the surgeries indicates whether these aims have been met. To make more decisive conclusions regarding the performance of 3D-printed medical devices in oral and maxillofacial surgery and musculoskeletal surgery, it is important to assess the long-term effects. Few papers assessed the long-term differences of using the 3D-printed devices and those that did had mixed results as to whether there were any differences in recovery times and outcomes. The other aspect that should be addressed in future studies is the appropriate choice of outcome measures. In the studies analysed in this review, there appears to be no evaluation of how the selected outcome measures match the purpose of the device.

Therapeutic devices such as implants, prosthetic limbs and orthotics offer the promise of revolutionising the medical industry because of their ability to be custom made for the patient, but most therapeutic devices are still early stage, with little research into their efficacy and effectiveness in a clinical setting. There were not enough therapeutic devices tested against a control group to find and evaluate repeated measures of success.

Most studies reviewed were poor quality, meaning that their study designs demonstrated levels of evidence for the 3D-printed device of 4 or 5, with no use of a control group. The 41.4% of papers that reported on individual case studies or provided only qualitative results did not clinically validate the devices that they reported on and therefore add little value to their fields. Few included follow-up or addressed the external validity, bias or confounding factors (see online supplementary material for full comparison of studies). The vast majority of papers reported positive outcomes for the patients, but few studies demonstrated clinically significant findings. Even the level 1 studies demonstrated limited external validity and rarely blinded participants or those measuring the outcomes to the intervention.

The RCTs all demonstrated statistically significant results in favour of using the 3D-printed device over the current clinical practice to which it was compared. However, one RCT[23] concluded that the improvement with 3D printing was too small to provide a clinical advantage, and none performed an analysis on the clinical relevance of these differences. The studies consistently used p values of <0.05

 

**Table 3** Critical appraisal of studies with the highest level of evidence from each medical field

| + | Yes |
|---|-----|
| - | No |
| ? | Unable to determine |
| x | Not applicable |

| | Category | de Farias | Ahrberg | D'Urso | Al-Ahmad | Ayoub | Goh | Van de Velde | Vercruyssen | Chareancholvanich | Chen | Du | Gan | Hendel | Maini | Merc | Shuang | Stephens | Wu | Yang | You | Zhang |
|---|---|---|---|---|---|---|---|---|---|---|---|---|---|---|---|---|---|---|---|---|---|---|
| Medical sector | | Neoplasms | Oral and Maxillofacial | Oral and Maxillofacial | Oral and Maxillofacial | Oral and Maxillofacial | Oral and Maxillofacial | Oral and Maxillofacial | Oral and Maxillofacial | Musculoskeletal | Musculoskeletal | Musculoskeletal | Musculoskeletal | Musculoskeletal | Musculoskeletal | Musculoskeletal | Musculoskeletal | Musculoskeletal | Musculoskeletal | Musculoskeletal | Musculoskeletal | Musculoskeletal |
| Date | | 2014 | 2015 | 1999 | 2013 | 2014 | 2015 | 2010 | 2014 | 2013 | 2015 | 2012 | 2015 | 2012 | 2016 | 2013 | 2016 | 2016 | 2011 | 2016 | 2016 | 2011 |
| 1. Clear hypothesis/aim/objective | Reporting | + | + | + | + | + | + | + | + | + | + | + | + | + | + | + | + | + | + | + | + | + |
| 2. Clear outcome measures | Reporting | + | + | + | + | + | + | + | + | + | + | + | + | + | + | + | + | + | + | + | + | + |
| 3. Patient characteristics described | Reporting | + | + | + | + | + | + | + | + | + | + | - | + | - | + | + | - | + | - | - | - | - |
| 4. Interventions clearly described | Reporting | + | + | + | + | + | + | + | + | + | + | - | + | + | + | + | + | + | + | + | + | + |
| 5. Distributions of confounders described | Reporting | + | + | + | x | - | - | - | - | - | + | + | + | - | - | + | + | - | - | - | + | + |
| 6. Findings clearly described | Reporting | + | + | + | + | + | + | + | + | + | + | + | + | + | + | + | + | + | + | + | + | + |
| 7. Estimates given of random variability | Reporting | + | + | + | + | + | + | + | + | + | + | + | + | + | + | + | + | + | + | + | + | + |
| 8. Adverse events reported | Reporting | + | - | - | - | - | - | + | + | + | + | - | + | + | + | + | + | - | + | + | + | - |
| 9. Patients lost to follow-up described | Reporting | x | x | x | + | x | + | + | + | + | + | + | x | - | x | x | + | + | + | x | + | + |
| 10. Probability values reported | Reporting | + | + | - | + | - | + | + | - | + | - | + | + | + | + | + | + | + | + | + | + | + |
| 11. Recruitment pool represents population | External validity | ? | ? | ? | ? | ? | ? | ? | ? | ? | ? | ? | ? | + | ? | + | ? | ? | + | ? | + | ? |
| 12. Participants represent population | External validity | ? | ? | ? | ? | ? | ? | ? | ? | ? | ? | ? | ? | + | ? | + | ? | ? | + | ? | + | ? |
| 13. Staff/places/facilities match standard treatment | External validity | + | + | + | + | + | + | + | + | + | + | + | + | + | + | + | ? | + | + | + | + | + |
| 14. Participants blinded to intervention | Internal validity - bias | + | + | - | + | - | - | - | - | - | + | - | - | - | - | - | - | + | - | - | - | - |
| 15. Those measuring outcomes blinded | Internal validity - bias | + | + | - | + | - | + | + | + | + | - | - | + | + | - | - | - | + | - | - | - | - |
| 16. Data dredging reported | Internal validity - bias | + | + | + | + | + | + | + | + | + | + | + | + | + | + | + | + | + | + | + | + | + |
| 17. Adjusted for different lengths of follow-up | Internal validity - bias | x | x | x | + | x | + | + | + | - | + | - | x | ? | x | x | ? | + | - | x | - | - |
| 18. Appropriate statistical tests | Internal validity - bias | + | + | + | + | + | + | + | + | + | + | + | + | + | + | + | + | + | + | + | + | + |
| 19. Reliable compliance with intervention | Internal validity - bias | + | + | + | + | + | + | + | + | + | + | + | + | + | + | + | + | + | + | + | + | + |
| 20. Accurate/reliable outcome measures | Internal validity - bias | + | + | + | + | + | + | + | + | + | + | + | + | + | + | + | + | + | + | + | + | + |
| 21. Groups recruited from same population | Internal validity - confounding | + | + | + | + | + | + | + | + | + | + | + | + | + | + | + | + | + | + | + | + | + |
| 22. Groups recruited over same timeframe | Internal validity - confounding | + | + | + | + | + | + | + | + | + | + | + | + | + | + | + | + | + | + | + | + | + |
| 23. Subjects randomised into intervention | Internal validity - confounding | + | + | + | + | + | + | + | + | + | + | + | + | + | + | + | + | + | + | + | + | + |
| 24. Randomised intervention concealed | Internal validity - confounding | + | + | ? | + | ? | ? | ? | ? | ? | + | ? | ? | - | ? | ? | + | + | ? | ? | ? | ? |
| 25. Adjustment for confounding | Internal validity - confounding | ? | + | - | + | - | - | - | - | - | + | - | - | - | - | - | + | - | - | - | + | + |
| 26. Losses to follow-up accounted for | Internal validity - confounding | x | x | x | + | x | + | + | + | + | + | + | x | ? | x | x | + | + | + | x | + | + |
| 27. Sufficient power | Power | + | + | + | + | + | + | + | + | + | + | + | + | + | + | + | + | + | + | + | + | + |

as their measure of device success, but taking into consideration the false discovery rate, it is likely that this will only show a real effect about 70% of the time under ideal testing conditions.[24] While the critical analysis in table 3 shows all level 1 studies as adequately powered, this is because the Quality Index bases its definition of 'adequately powered' on whether a difference due to chance is less than 5%. Therefore, given that all level 1 studies had 40 or fewer participants in each group, many of these studies are likely to be underpowered.[24] Considering the possibility of underpowered studies, the publication bias towards positive trials and the fact that all RCTs in this review were single-site trials, the likelihood that the results shown in these papers reflect a real effect is lower than suggested by the reviewed studies.[25] Until independent groups start validating 3D-printed devices, it is also difficult to avoid researcher bias.

Most 3D-printed device studies that were excluded from this review had uses in medical training or as moulds for manufacturing and therefore did not meet the inclusion requirements. A few other studies were excluded because they had not been translated into English.

Previous reviews have documented the uses of 3D printing for developing patient-specific medical devices. However, there has been limited research into assessing the efficacy and effectiveness of these devices. This rigorous systematic review design is the first to compare 3D-printed devices across all medical fields and assess their efficacy and effectiveness, describing key benefits that have been found from using 3D-printed devices clinically. The progress made in different medical fields is compared using validated quality assessment and clinical level of evidence tools. It demonstrates that the fields of oral and maxillofacial surgery and the musculoskeletal system are leading the way in validating 3D-printed devices for clinical use. Multiple high-quality studies have been performed on surgical guides for maxillofacial, hip and knee surgeries. This growing body of comparable high-quality research sets an example for other fields to emulate in order to demonstrate the efficacy and effectiveness required to integrate these 3D-printed devices into clinical practice.

A critical appraisal of the efficacy and effectiveness of 3D-printed devices across medical fields provides clinicians with an evidence-based approach to determine the applicability of 3D printing within their field. It also gives researchers an overview of areas that require further research and validation. It encourages investigators to discover what methods have been successfully validated in other medical fields and promotes potential collaborations between fields.

Three-dimensional printing provides a way of customising devices to improve patient outcomes. Available techniques and materials will increase, as 3D printing technology continues to be developed. There is therefore a growing need for validation of new devices, materials and techniques to ensure best patient outcomes. Much research has already gone into developing 3D-printed devices for medical purposes. However, this drive for new technology development has not yet been matched with a drive for critical appraisal of the devices to demonstrate their efficacy and effectiveness. Even the fields that are leading the way in critically evaluating new 3D-printed devices will be required to increase their output as 3D printing continues to grow in popularity and functionality.

The early research covered by this review shows that 3D printing can be valuable for use in medicine. The next important step to take is growing the body of research that focuses on validating 3D-printed devices. All fields require more rigorous and long-term assessments into the efficacy, effectiveness and safety of 3D-printed devices before they are introduced into standard clinical practice. The study is limited to a critical appraisal of individual studies, rather than a meta-analysis, because of the breadth of uses for anatomical models, surgical guides and therapeutic devices and the lack of comparable hypotheses. Funders can take an active role in promoting not only early technological development but also the subsequent clinical trials. Demonstration of clinical efficacy, effectiveness and device safety will become increasingly important as higher risk 3D-printed devices are developed and unconventional manufacturers, such as hospitals and clinics, incorporate 3D printing of patient-specific medical devices into standard clinical practice.

Fields such as oral and maxillofacial surgery, who were early in the uptake of 3D printing, are beginning to stabilise, with a steady number of studies being published each year, but other fields, such as musculoskeletal and circulatory systems, are more recently gaining traction, with increasing numbers of studies performed each year. 3D-printed drug delivery devices and biological 3D printing technologies for printing tissue show huge promise but have not been clinically trialled and so are not included here.[26] This is likely to change within the near future. It is therefore recommended that, in this fast-growing and dynamic environment, this review is updated every few years.

## CONCLUSION

This review demonstrates that 3D printing is already being used to develop a broad range of medical devices with clinically effective results. The medical fields of oral and maxillofacial surgery and the musculoskeletal system are leading the way in validating the efficacy and effectiveness of 3D-printed devices and have found that 3D-printed anatomical models and surgical guides are reducing operating times and increasing surgical accuracy. However, the efficacy and effectiveness of 3D-printed devices remains undetermined for the majority of medical fields. 3D-printed devices can have an important role to play in healthcare, but more rigorous and long-term assessments are needed to determine if 3D-printed devices are clinically relevant before these devices can become part of standard clinical practice.

**Contributors** LED, JHMB and MST designed the review. LED and JHMB collected and analysed the data with input from MST. LED drafted the manuscript. All authors contributed to subsequent drafts and approved the final version of the manuscript.

**Funding** The researchers are financially supported by the General Sir John Monash Foundation and the Wellcome Trust (103383/B/13/Z).

**Competing interests** None declared.

**Provenance and peer review** Not commissioned; externally peer reviewed.

**Data sharing statement** All additional data are available in the supplementary material.

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
