## [Reviewer comments · BMJ Open]

ARTICLE DETAILS

TITLE (PROVISIONAL)	Clinical efficacy and effectiveness of 3D printing: A systematic review
AUTHORS	Diment, Laura; Thompson, Mark; Bergmann, Jeroen

VERSION 1 – REVIEW

REVIEWER	Amir A Zadpoor Delft University of Technology No Competing Interest
REVIEW RETURNED	06-Apr-2017

GENERAL COMMENTS	The topic of this review is timely and important. The methodology followed by the authors is generally good and suitable for the topic at hand. I have one major concern throughout the paper: in my opinion, the results of the clinical studies in the oral/maxillofacial and musculoskeletal areas are not adequately presented and discussed. I think the authors emphasize what is not covered in the literature a little more than what is already available in the literature. I suggest that the authors present an overview of the findings of the included clinical trials in the Results part of the paper and explain what has been consistently found in the included studies (and what has not). I feel that the same approach should be also continued in the discussion. The discussion should discuss the findings of the included studies. Why do these studies find what they find? What could be the mechanistic reasons for those findings? What are the exact research questions in oral/maxillofacial and musculoskeletal research that need to be addressed to make more decisive conclusions regarding the performance of AM medical devices? I guess what I am trying to say is that given the fact that most clinical studies are in oral/maxillofacial and musculoskeletal research, the review should focus on the findings of those studies and present an in-depth analysis of their findings. The need for clinical studies in other areas could be mentioned as well but as a side note and not as the most important conclusion of the study. The abstract and conclusions of the study should be then modified to reflect the findings and limitations of the included studies (i.e. the studies in the oral/maxillofacial and musculoskeletal fields).
---

REVIEWER	Kenneth R. Foster Dept of Bioengineering Univ of PA Phila PA 19106 USA No Competing Interest
REVIEW RETURNED	17-Apr-2017

GENERAL COMMENTS	Very nice paper and much needed analysis, will go a long way to clarify the (generally weak) state of assessment of safety and effectiveness of 3D printing devices. Some comments for minor revision 1. Intro. FDA requirements for testing. Most approved AM devices (maxillofacial implants etc) are FDA Class II. For Class II devices FDA does *not* require RCTs demonstrating safety and effectiveness but rather a demonstration that that devices are "substantially equivalent" to already approved devices. The FDA guidance (ref 9) does point out that AM can raise safety and efficacy issues different than those in equivalent non-AM devices which would require further testing, but those may be addressed by in vivo (animal) studies, not RCTs with human patients. Also, in a passage shortly after, mention of how nontraditional manufacturers "such as hospitals and clinics" must "adhere" to FDA regulations is not necessarily pertinent. A medical device firm that makes bespoke implants using patient-specific data would certainly be subject to FDA manufacturing controls but FDA premarket approval requirements, design controls, etc would not necessarily apply to healthcare centers. 2. Conclusion para - pointing out need for effectiveness and efficacy testing is OK but should also call for need for safety testing. The Oct 2014 FDA Workshop on 3D printing had a lot of discussion about safety issues (sterilization), design issues, material equivalence to conventionally manufactured devices, but little if any call for large scale RCTs. 3. Pervasive misuse of statistics in Table 2 (p-values). (a) The p value is not a measure of effect size. Please indicate the effect size in entries where only p values are presently given. For example in "fewer abnormal self-reported changes in lower lip sensation (p<0.05 at 1 week)" - how large was the effect? (b) A comparison with p<0.05 should not be referred to as "significant" without qualification (ambiguous) but "statistically significant" or "medically significant". For example does the statement "The test group showed significantly lower values of internal fit at the centro-occlusal site" refer to a "statistically significant" or "medically significant" difference? Please clarify. (c) the p value derived from null hypothesis significance testing is basically useless - it indicates the conditional probability of the observed or greater difference between two groups assuming that the null hypothesis, *not* the probability that the difference arises from a real effect. For low-power studies (e.g. virtually all of those considered in this review) the probability of false discovery with p < 0.05 is very high.(Colquhoun D. 2014 An the false discovery rate and misinterpretation of p-values. R. Soc. open 10.1098/rsos.140216)). My "take home" lesson is that the RCTs discussed here, which are uniformly small and with low statistical power, are even less persuasive of a real clinical benefit compared to conventionally manufactured devices than the authors conclude - is that correct?
--

	Colquhoun's analysis (as well as similar comments by many others, including the famous paper by Ionnaidis on false discovery) might be worth mentioning in this context. ref: Foster KR "3-Dimensional Printing in Medicine: Hype, Hope, and The Challenge of Personalized Medicine", in Philosophy and Engineering: Exploring Boundaries, Expanding Connections, B. Newberry, D. Michelfelder, Q. Zhu, Springer, Volume 26 of the series Philosophy of Engineering and Technology pp 211-228 (2016).
--	---

REVIEWER	Abby Paterson Loughborough University, UK
REVIEW RETURNED	16-May-2017

GENERAL COMMENTS	I would like to commend the authors; this is a highly relevant area of research and has scope to generate significant impact as a result. It will be a very useful reference for many in academia, clinical practice, industry, and beyond. The reviewer would appreciate the following corrections:  - Spelling mistake on P3, line 35/56 ('...that did asses') The aim and objectives were identifiable, but could be labelled more clearly.  - Please explain in more detail how the references were managed; was a reference manager (e.g. Refworks, Mendeley) used? Or Microsoft Excel to deal with additional fields of information? Please clarify. - The main body of the text refers to 'hypotheses', but many of the studies may have an aim and objectives. You make reference to aims and objectives in Table 3, but it would be good to see addition of 'aim and objectives' to the main body of the text, otherwise some may feel that only papers with hypotheses were considered. - please provide more information on some of the forms of validation. Some aspects could be considered subjective (e.g. 'simple and intuitive to interpret), and down to the interpretation of the reader, so how was this accounted for? The paper mentions double-blind review was used on the titles; was that true for the other criteria? - how were criteria categorised under 'Yes', 'No' 'Unable to Determine' categories in Table 3? Is this open to interpretation? How was that compensated? - On page 9, it states that the majority of papers were 'poor quality'; please explicitly state what was interpreted as poor quality. Some of the papers listed are considered high quality for their intended purpose of exploration and validation of a specific methodology/process, so please define exactly what constitutes 'poor quality' in the context of this study (even if it is a simple reference back to the inclusion criteria as ideal characteristics). P16 shows the keywords used in the different databases, yet the words and the quantity used varies. Please explain more thoroughly as to why this is.
--

VERSION 1 – AUTHOR RESPONSE

Response to comments from Reviewer 1:

Comment: "The topic of this review is timely and important. The methodology followed by the authors is generally good and suitable for the topic at hand. I have one major concern throughout the paper: in my opinion, the results of the clinical studies in the oral/maxillofacial and musculoskeletal areas are not adequately presented and discussed. I think the authors emphasize what is not covered in the literature a little more than what is already available in the literature. I suggest that the authors present an overview of the findings of the included clinical trials in the Results part of the paper and explain what has been consistently found in the included studies (and what has not). I feel that the same approach should be also continued in the discussion. The discussion should discuss the findings of the included studies. Why do these studies find what they find? What could be the mechanistic reasons for those findings? What are the exact research questions in oral/maxillofacial and musculoskeletal research that need to be addressed to make more decisive conclusions regarding the performance of AM medical devices? I guess what I am trying to say is that given the fact that most clinical studies are in oral/maxillofacial and musculoskeletal research, the review should focus on the findings of those studies and present an in-depth analysis of their findings. The need for clinical studies in other areas could be mentioned as well but as a side note and not as the most important conclusion of the study. The abstract and conclusions of the study should be then modified to reflect the findings and limitations of the included studies (i.e. the studies in the oral/maxillofacial and musculoskeletal fields)."

Response: A fuller summary of the outcomes of each paper has been added to Table 2, and a summary of the main findings from the key papers in Oral and Maxillofacial, and Musculoskeletal Surgery is now provided in the Results section with an overview of what has been consistently found in the studies in regards to type of device, success measures used and study outcomes. The Discussion now contains an in-depth analysis of the findings from these studies and the possible reasons for these findings, as well as discussing the research questions required in these fields to make more decisive conclusions regarding the performance of 3D printed devices.

The abstract and conclusions of the study have been modified to reflect the findings and limitations of the included Level 1 studies in the Oral and Maxillofacial, and Musculoskeletal fields.

Response to comments from Reviewer 2:

"Very nice paper and much needed analysis, will go a long way to clarify the (generally weak) state of assessment of safety and effectiveness of 3D printing devices.

Comment 1. Intro. FDA requirements for testing. Most approved AM devices (maxillofacial implants etc) are FDA Class II. For Class II devices FDA does *not* require RCTs demonstrating safety and effectiveness but rather a demonstration that that devices are "substantially equivalent* to already approved devices. The FDA guidance (ref 9) does point out that AM can raise safety and efficacy issues different than those in equivalent non-AM devices which would require further testing, but those may be addressed by in vivo (animal) studies, not RCTs with human patients. Also, in a passage shortly after, mention of how nontraditional manufacturers "such as hospitals and clinics" must "adhere" to FDA regulations is not necessarily pertinent. A medical device firm that makes bespoke implants using patient-specific data would certainly be subject to FDA manufacturing controls but FDA premarket approval requirements, design controls, etc would not necessarily apply to healthcare centers."

Response: The Introduction and Discussion have been updated to make sure the statements on FDA regulations and the responsibilities of hospitals and clinics are correct and in line with the information the reviewer provided. Given that the regulatory requirements are not the primary focus of the paper, these sections have been kept brief.

Comment 2. "Conclusion para - pointing out need for effectiveness and efficacy testing is OK but should also call for need for safety testing. The Oct 2014 FDA Workshop on 3D printing had a lot of discussion about safety issues (sterilization), design issues, material equivalence to conventionally manufactured devices, but little if any call for large scale RCTs."

Response: Mention of the need for safety testing has been added to the Discussion. The authors recognise the importance of safety testing but it is outside the focus of this study and does not match up with the study aim. It therefore it does not belong in the concluding paragraph.

Comment 3. "Pervasive misuse of statistics in Table 2 (p-values). (a) The p value is not a measure of effect size. Please indicate the effect size in entries where only p values are presently given. For example in "fewer abnormal self-reported changes in lower lip sensation ($p < 0.05$ at 1 week)" - how large was the effect? (b) A comparison with $p < 0.05$ should not be referred to as "significant" without qualification (ambiguous) but "statistically significant" or "medically significant". For example does the statement "The test group showed significantly lower values of internal fit at the centro-occlusal site" refer to a "statistically significant" or "medically significant" difference? Please clarify. (c) the p value derived from null hypothesis significance testing is basically useless - it indicates the conditional probability of the observed or greater difference between two groups assuming that the null hypothesis, *not* the probability that the difference arises from a real effect. For low-power studies (e.g. virtually all of those considered in this review) the probability of false discovery with $p < 0.05$ is very high. (Colquhoun D. 2014 An the false discovery rate and misinterpretation of p-values. R. Soc. open 10.1098/rsos.140216)). My "take home" lesson is that the RCTs discussed here, which are uniformly small and with low statistical power, are even less persuasive of a real clinical benefit compared to conventionally manufactured devices than the authors conclude - is that correct? Colquhoun's analysis (as well as similar comments by many others, including the famous paper by Ionnidis on false discovery) might be worth mentioning in this context."

Response: Effect size has been added to all results, and clarification of the word "significant" has been added where required. It is correct that the p-value derived from null hypothesis significance testing does not demonstrate proof of a clinical benefit, but it is the primary measure used throughout the studies to demonstrate device success. It is also true that a real clinical benefit compared to conventionally manufactured devices has not been adequately shown by these papers. The authors have added a section to the Discussion that discusses the probability of false discovery using $p < 0.05$, the effect of underpowered studies and the additional questions that should be answered by future studies to better assess the clinical benefits of these devices. Both Colquhoun and Ionnidis have been referenced.

Comment 4. "ref: Foster KR "3-Dimensional Printing in Medicine: Hype, Hope, and The Challenge of Personalized Medicine", in Philosophy and Engineering: Exploring Boundaries, Expanding Connections, B. Newberry, D. Michelfelder, Q. Zhu, Springer, Volume 26 of the series Philosophy of Engineering and Technology pp 211-228 (2016)."

Response: The authors are aware that many papers and book chapters have been published on 3D printing in medicine, discussing the possibilities and challenges. However, the authors chose to reference only systematic reviews, and therefore this suggested paper has not been referenced. The term "systematic" was added to the relevant section in the Introduction to explain that the referenced papers are all systematic reviews.

Response to comments from Reviewer 3.

"I would like to commend the authors; this is a highly relevant area of research and has scope to generate significant impact as a result. It will be a very useful reference for many in academia, clinical practice, industry, and beyond.

Comment 1. Spelling mistake on P3, line 35/56 ('...that did asses')"

Response: The spelling mistake has been fixed.

Comment 2. "The aim and objectives were identifiable, but could be labelled more clearly."

Response: The terminology in the aim has modified to make it clearer.

Comment 3. "Please explain in more detail how the references were managed; was a reference manager (e.g. Refworks, Mendeley) used? Or Microsoft Excel to deal with additional fields of information? Please clarify."

Response: Mendeley was used and has been added to the methods.

Comment 4. "The main body of the text refers to 'hypotheses', but many of the studies may have an aim and objectives. You make reference to aims and objectives in Table 3, but it would be good to see addition of 'aim and objectives' to the main body of the text, otherwise some may feel that only papers with hypotheses were considered."

Response: 'Aim and objective' have been added to 'hypothesis' in the main body of the text.

Comment 5. "Please provide more information on some of the forms of validation. Some aspects could be considered subjective (e.g. 'simple and intuitive to interpret), and down to the interpretation of the reader, so how was this accounted for? The paper mentions double-blind review was used on the titles; was that true for the other criteria? How were criteria categorised under 'Yes', 'No' 'Unable to Determine' categories in Table 3? Is this open to interpretation? How was that compensated?"

Response: The information on interpretation and how the subjective nature of some of the validation questions was accounted for has been added to the Methods section: The Quality Index gives explanations to clarify how subjective information should be categorised, and where there were further uncertainties, the primary author consulted the other two authors and an agreement was reached.

Comment 6. "On page 9, it states that the majority of papers were 'poor quality'; please explicitly state what was interpreted as poor quality. Some of the papers listed are considered high quality for their intended purpose of exploration and validation of a specific methodology/process, so please define exactly what constitutes 'poor quality' in the context of this study (even if it is a simple reference back to the inclusion criteria as ideal characteristics)."

Response: The Discussion has been updated to explain what is meant by 'poor quality' studies.

Comment 7. "P16 shows the keywords used in the different databases, yet the words and the quantity used varies. Please explain more thoroughly as to why this is."

Response: This is because of the word and character limit search restrictions in IEEE Xplore and Google Scholar. The explanation at the start of the Methods section has been adjusted for clarification.

VERSION 2 – REVIEW

REVIEWER	Kenneth R. Foster Univ of PA. Philadelphia PA USA
REVIEW RETURNED	24-Aug-2017
GENERAL COMMENTS	good paper!